# Densely connected normalizing flows

**Matej Grcić, Ivan Grubišić and Siniša Šegvić**
Faculty of Electrical Engineering and Computing
University of Zagreb
`matej.grcic@fer.hr ivan.grubisic@fer.hr sinisa.segvic@fer.hr`

## Abstract

Normalizing flows are bijective mappings between inputs and latent representations with a fully factorized distribution. They are very attractive due to exact likelihood evaluation and efficient sampling. However, their effective capacity is often insufficient since the bijectivity constraint limits the model width. We address this issue by incrementally padding intermediate representations with noise. We precondition the noise in accordance with previous invertible units, which we describe as cross-unit coupling. Our invertible glow-like modules increase the model expressivity by fusing a densely connected block with Nyström self-attention. We refer to our architecture as DenseFlow since both cross-unit and intra-module couplings rely on dense connectivity. Experiments show significant improvements due to the proposed contributions and reveal state-of-the-art density estimation under moderate computing budgets.[1]

## 1 Introduction

One of the main tasks of modern artificial intelligence is to generate images, audio waveforms, and natural-language symbols. To achieve the desired goal, the current state of the art uses deep compositions of non-linear transformations [1, 2] known as *deep generative models* [3, 4, 5, 6, 7]. Formally, deep generative models estimate an unknown data distribution $p_D$ given by a set of i.i.d. samples $D = \{\boldsymbol{x}_1, ..., \boldsymbol{x}_n\}$. The data distribution is approximated with a model distribution $p_\theta$ defined by the architecture of the model and a set of parameters $\theta$. While the architecture is usually handcrafted, the set of parameters $\theta$ is obtained by optimizing the likelihood across the training distribution $p_D$:

$$\theta^* = \underset{\theta \in \Theta}{\operatorname{argmin}} \, \mathbb{E}_{\boldsymbol{x} \sim p_D}[-\ln p_\theta(\boldsymbol{x})]. \tag{1}$$

Properties of the model (e.g. efficient sampling, ability to evaluate likelihood etc.) directly depend on the definition of $p_\theta(\boldsymbol{x})$, or decision to avoid it. Early approaches consider unnormalized distribution [3] which usually requires MCMC-based sample generation [8, 9, 10] with long mixing times. Alternatively, the distribution can be autoregressively factorized [7, 11], which allows likelihood estimation and powerful but slow sample generation. VAEs [4] use a factorized variational approximation of the latent representation, which allows to learn an autoencoder by optimizing a lower bound of the likelihood. Diffussion models [12, 13, 14] learn to reverse a diffusion process, which is a fixed Markov chain that gradually adds noise to the data in the opposite direction of sampling until the signal is destroyed. Generative adversarial networks [5] mimic the dataset samples by competing in a minimax game. This allows to efficiently produce high quality samples [15], which however often do not span the entire training distribution support [16]. Additionally, the inability to "invert" the generation process in any meaningful way implies inability to evaluate the likelihood.

Contrary to previous approaches, normalizing flows [6, 17, 18] model the likelihood using a bijective mapping to a predefined latent distribution $p(\boldsymbol{z})$, typically a multivariate Gaussian. Given the bijection

---

[1]Code available at: `https://github.com/matejgrcic/DenseFlow`

$f_\theta$, the likelihood is defined using the change of variables formula:

$$p_\theta(\boldsymbol{x}) = p(\boldsymbol{z}) \left| \det \frac{\partial \boldsymbol{z}}{\partial \boldsymbol{x}} \right|, \quad \boldsymbol{z} = f_\theta(\boldsymbol{x}). \tag{2}$$

This approach requires computation of the Jacobian determinant ($\det \frac{\partial \boldsymbol{z}}{\partial \boldsymbol{x}}$). Therefore, during the construction of bijective transformations, a great emphasis is placed on tractable determinant computation and efficient inverse computation [18, 19]. Due to these constraints, invertible transformations require more parameters to achieve a similar capacity compared to standard NN building blocks [20]. Still, modeling $p_\theta(\boldsymbol{x})$ using bijective formulation enables exact likelihood evaluation and efficient sample generation, which makes this approach convenient for various downstream tasks [21, 22, 23].

The bijective formulation (2) implies that the input and the latent representation have the same dimensionality. Typically, convolutional units of normalizing-flow approaches [18] internally inflate the dimensionality of the input, extract useful features, and then compress them back to the original dimensionality. Unfortunately, the capacity of such transformations is limited by input dimensionality [24]. This issue can be addressed by expressing the model as a sequence of bijective transformations [18]. However, increasing the depth alone is a suboptimal approach to improve capacity of a deep model [25]. Recent works propose to widen the flow by increasing the input dimensionality [24, 26]. We propose an effective development of that idea which further improves the performance while relaxing computational requirements.

We increase the expressiveness of normalizing flows by incremental augmentation of intermediate latent representations with Gaussian noise. The proposed cross-unit coupling applies an affine transformation to the noise, where the scaling and translation are computed from a set of previous intermediate representations. In addition, we improve intra-module coupling by proposing a transformation which fuses the global spatial context with local correlations. The proposed image-oriented architecture improves expressiveness and computational efficiency. Our models set the new state-of-the-art result in likelihood evaluation on ImageNet32 and ImageNet64.

## 2 Densely connected normalizing flows

We present a recursive view on normalizing flows and propose improvements based on incremental augmentation of latent representations, and densely connected coupling modules paired with self-attention. The improved framework is then used to develop an image-oriented architecture, which we evaluate in the experimental section.

### 2.1 Normalizing flows with cross-unit coupling

Normalizing flows (NF) achieve their expressiveness by stacking multiple invertible transformations [18]. We illustrate this with the scheme (3) where each two consecutive latent variables $\boldsymbol{z}_{i-1}$ and $\boldsymbol{z}_i$ are connected via a dedicated flow unit $f_i$. Each flow unit $f_i$ is a bijective transformation with parameters $\theta_i$ which we omit to keep notation uncluttered. The variable $\boldsymbol{z}_0$ is typically the input $\boldsymbol{x}$ drawn from the data distribution $p_D(\boldsymbol{x})$.

$$\boldsymbol{z}_0 \xleftrightarrow{f_1} \boldsymbol{z}_1 \xleftrightarrow{f_2} \boldsymbol{z}_2 \xleftrightarrow{f_3} \cdots \xleftrightarrow{f_{i-1}} \boldsymbol{z}_i \xleftrightarrow{f_i} \cdots \xleftrightarrow{f_K} \boldsymbol{z}_K, \quad \boldsymbol{z}_K \sim \mathcal{N}(0, \mathrm{I}). \tag{3}$$

Following the change of variables formula, log likelihoods of consecutive random variables $\boldsymbol{z}_i$ and $\boldsymbol{z}_{i+1}$ can be related through the Jacobian of the corresponding transformation $\mathbf{J}_{f_{i+1}}$ [18]:

$$\ln p(\boldsymbol{z}_i) = \ln p(\boldsymbol{z}_{i+1}) + \ln |\det \mathbf{J}_{f_{i+1}}|. \tag{4}$$

This relation can be seen as a recursion. The term $\ln p(\boldsymbol{z}_{i+1})$ can be recursively replaced either with another instance of (4) or evaluated under the latent distribution, which marks the termination step. This setup is characteristic for most contemporary architectures [17, 18, 19, 27].

The standard NF formulation can be expanded by augmenting the input by a noise variable $\boldsymbol{e}_i$ [24, 26]. The noise $\boldsymbol{e}_i$ subjects to some known distribution $p^*(\boldsymbol{e}_i)$, e.g. a multivariate Gaussian. We further improve this approach by incrementally concatenating noise to each intermediate latent representation $\boldsymbol{z}_i$. A tractable formulation of this idea can be obtained by computing the lower bound of the likelihood $p(\boldsymbol{z}_i)$ through Monte Carlo sampling of $\boldsymbol{e}_i$:

$$\ln p(\boldsymbol{z}_i) \geq \mathbb{E}_{\boldsymbol{e}_i \sim p^*(\boldsymbol{e})} \left[ \ln p(\boldsymbol{z}_i, \boldsymbol{e}_i) - \ln p^*(\boldsymbol{e}_i) \right]. \tag{5}$$

The learned joint distribution $p(\boldsymbol{z}_i, \boldsymbol{e}_i)$ approximates the product of the target distributions $p^*(\boldsymbol{z}_i)$ and $p^*(\boldsymbol{e}_i)$, which is explained in more detail in Appendix D. We transform the introduced noise $\boldsymbol{e}_i$ with element-wise affine transformation. Parameters of this transformation are computed by a learned non-linear transformation $g_i(\boldsymbol{z}_{<i})$ of previous representations $\boldsymbol{z}_{<i} = [\boldsymbol{z}_0, ..., \boldsymbol{z}_{i-1}]$. The resulting layer $h_i$ can be defined as:

$$\boldsymbol{z}_i^{(\text{aug})} = h_i(\boldsymbol{z}_i, \boldsymbol{e}_i, \boldsymbol{z}_{<i}) = [\boldsymbol{z}_i, \boldsymbol{\sigma} \odot \boldsymbol{e}_i + \boldsymbol{\mu}], \quad (\boldsymbol{\mu}, \boldsymbol{\sigma}) = g_i(\boldsymbol{z}_{<i}). \tag{6}$$

Square brackets $[\cdot, \cdot]$ denote concatenation along the features dimension. In order to compute the likelihood for $(\boldsymbol{z}_i, \boldsymbol{e}_i)$, we need the determinant of the jacobian

$$\frac{\partial \boldsymbol{z}_i^{(\text{aug})}}{\partial [\boldsymbol{z}_i, \boldsymbol{e}_i]} = \begin{bmatrix} \mathrm{I} & 0 \\ 0 & \mathrm{diag}(\boldsymbol{\sigma}) \end{bmatrix}. \tag{7}$$

Now we can express $p(\boldsymbol{z}_i, \boldsymbol{e}_i)$ in terms of $p(\boldsymbol{z}_i^{(\text{aug})})$ according to (4):

$$\ln p(\boldsymbol{z}_i, \boldsymbol{e}_i) = \ln p(\boldsymbol{z}_i^{(\text{aug})}) + \ln |\det \mathrm{diag}(\boldsymbol{\sigma})|. \tag{8}$$

We join equations (5) and (8) into a single step:

$$\ln p(\boldsymbol{z}_i) \geq \mathbb{E}_{\boldsymbol{e}_i \sim p^*(\boldsymbol{e}_i)}[\ln p(\boldsymbol{z}_i^{(\text{aug})}) - \ln p^*(\boldsymbol{e}_i) + \ln |\det \mathrm{diag}(\boldsymbol{\sigma})|]. \tag{9}$$

We refer the transformation $h_i$ as *cross-unit coupling* since it acts as an affine coupling layer [17] over a group of previous invertible units. The latent part of the input tensor is propagated without change, while the noise part is linearly transformed. The noise transformation can be viewed as reparametrization of the distribution from which we sample the noise [4]. Note that we can conveniently recover $\boldsymbol{z}_i$ from $\boldsymbol{z}_i^{(\text{aug})}$ by removing the noise dimensions. This step is performed during model sampling.

Fig. 1 compares the standard normalizing flow (a) normalizing flow with input augmentation [24] (b) and the proposed densely connected incremental augmentation with cross-unit coupling (c). Each flow unit $f_i^{\text{DF}}$ consists of several invertible modules $m_{i,j}$ and cross-unit coupling $h_i$. The main novelty of our architecture is that each flow unit $f_{i+1}^{\text{DF}}$ increases the dimensionality with respect to its predecessor $f_i^{\text{DF}}$. Cross-unit coupling $h_i$ augments the latent variable $\boldsymbol{z}_i$ with affinely transformed noise $\boldsymbol{e}_i$. Parameters of the affine noise transformation are obtained by an arbitrary function $g_i$ which accepts all previous variables $\boldsymbol{z}_{<i}$. Note that reversing the direction does not require evaluating $g_i$ since we are only interested in the value of $\boldsymbol{z}_i$. For further clarification, we show the likelihood computation for the extended framework.

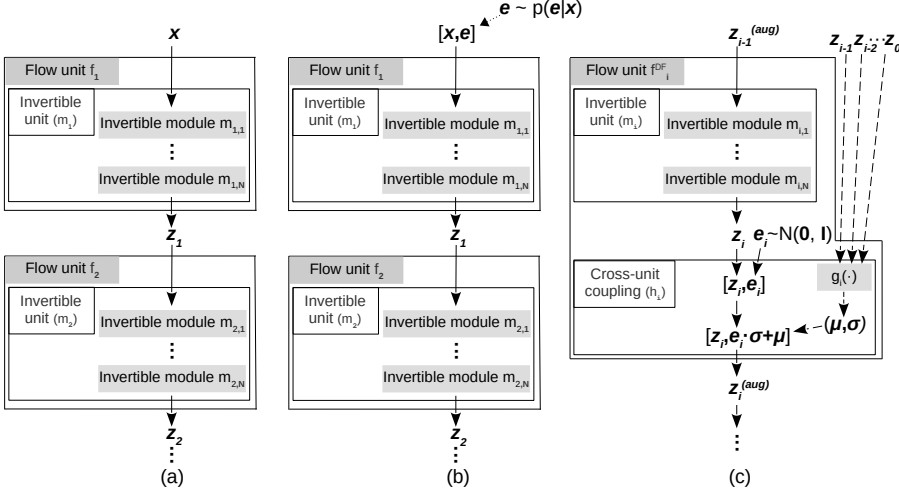

Figure 1: Standard normalizing flow [17, 18] (a), normalizing flow with augmented input [24] (b), and the proposed incremental augmentation with cross-unit coupling (c). Unlike (b) which adds noise only to the input, (c) adds noise to the output of every unit except the last.

**Example 1 (Likelihood computation)** *Let $m_1$ and $m_2$ be the bijective mappings from $\boldsymbol{z}_0$ to $\boldsymbol{z}_1$ and $\boldsymbol{z}_1^{(aug)}$ to $\boldsymbol{z}_2$, respectively. Let $h_1$ be the cross-unit coupling from $\boldsymbol{z}_1$ to $\boldsymbol{z}_1^{(aug)}$, $\boldsymbol{z}_1^{(aug)} = [\boldsymbol{z}_1, \boldsymbol{\sigma} \odot \boldsymbol{e}_1 + \boldsymbol{\mu}]$. Assume $\boldsymbol{\sigma}$ and $\boldsymbol{\mu}$ are computed by any non-invertible neural network $g_1$. The network accepts $\boldsymbol{z}_0$ as the input. We calculate log likelihood of the input $\boldsymbol{z}_0$ according to the following sequence of equations: [transformation, cross-unit coupling, transformation, termination].*

$$\ln p(\boldsymbol{z}_0) = \ln p(\boldsymbol{z}_1) + \ln|\det J_{f_1}|, \tag{10}$$

$$\ln p(\boldsymbol{z}_1) \geq \mathbb{E}_{\boldsymbol{e}_1 \sim p^*(\boldsymbol{e}_1)}[\ln p(\boldsymbol{z}_1^{(aug)}) - \ln p(\boldsymbol{e}_1) + \ln|\det \mathrm{diag}(\boldsymbol{\sigma})|], \quad (\boldsymbol{\sigma}, \boldsymbol{\mu}) = g_1(\boldsymbol{z}_0), \tag{11}$$

$$\ln p(\boldsymbol{z}_1^{(aug)}) = \ln p(\boldsymbol{z}_2) + \ln|\det J_{f_2}|, \tag{12}$$

$$\ln p(\boldsymbol{z}_2) = \ln \mathcal{N}(\boldsymbol{z}_2; 0, \mathrm{I}). \tag{13}$$

We approximate the expectation using MC sampling with a single sample during training and a few hundreds of samples during evaluation to reduce the variance of the likelihood. Note however that our architecture generates samples with a single pass since the inverse does not require MC sampling.

We repeatedly apply the cross-unit coupling $h_i$ throughout the architecture to achieve incremental augmentation of intermediate latent representations. Consequently, the data distribution is modeled in a latent space of higher dimensionality than the input space [24, 26]. This enables better alignment of the final latent representation with the NF prior. We materialize the proposed expansion of the normalizing flow framework by developing an image-oriented architecture which we call *DenseFlow*.

## 2.2 Image-oriented invertible module

We propose a glow-like invertible module (also known as step of flow [19]) consisting of activation normalization, $1 \times 1$ convolution and intra-module affine coupling layer. The attribute "intra-module" emphasizes distinction with respect to cross-unit coupling. Different than in the original glow design, our coupling network leverages advanced transformations based on dense connectivity and fast self-attention. All three layers are designed to capture complex data dependencies while keeping tractable Jacobians and efficient inverse computation. For completeness, we start by reviewing elements of the original glow module [19].

**ActNorm** [19] is an invertible substitute for batch normalization [30]. It performs affine transformation with per-channel scale and bias parameters:

$$\boldsymbol{y}_{i,j} = \boldsymbol{s} \odot \boldsymbol{x}_{i,j} + \boldsymbol{b}. \tag{14}$$

Scale and bias are calculated as the variance and mean of the initial minibatch.

**Invertible $1 \times 1$ Convolution** is a generalization of channel permutation [19]. Convolutions with $1 \times 1$ kernel are not invertible by construction. Instead, a combination of orthogonal initialization and the loss function keeps the kernel inverse numerically stable. The normalizing flow loss maximizes $\ln|\det \mathbf{J}_f|$ which is equivalent to maximizing $\sum_i \ln|\lambda_i|$, where $\lambda_i$ are eigenvalues of the Jacobian. Maintaining a relatively large amplitude of the eigenvalues ensures a stable inversion. The Jacobian of this transformation can be efficiently computed by LU-decomposition [19].

**Affine Coupling** [18] splits the input tensor $\boldsymbol{x}$ channel-wise into two halves $\boldsymbol{x}_1$ and $\boldsymbol{x}_2$. The first half is propagated without changes, while the second half is linearly transformed (15). The parameters of the linear transformation are calculated from the first half. Finally, the two results are concatenated as shown in Fig. 2.

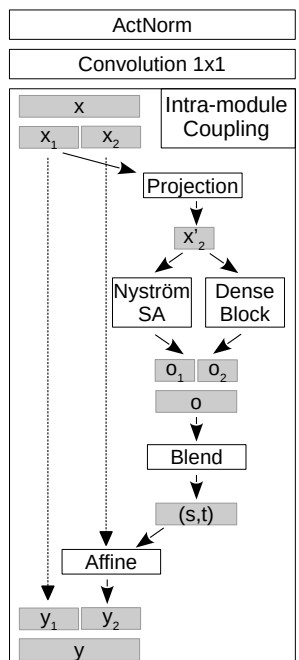

Figure 2: A glow-like module $m_{i,j}$ consist of ActNorm, 1x1 convolution and intra-module affine coupling. The proposed intra-module coupling fuses the global context recovered by fast self-attention [28] and local correlations extracted by densely connected convolutions [29].

$$\boldsymbol{y}_1 = \boldsymbol{x}_1, \quad \boldsymbol{y}_2 = \boldsymbol{s} \odot \boldsymbol{x}_2 + \boldsymbol{t}, \quad (\boldsymbol{s}, \boldsymbol{t}) = coupling\_net(\boldsymbol{x}_1). \tag{15}$$

Parameters $s$ and $t$ are calculated using a trainable network which is typically implemented as a residual block [18]. However, this setting can only capture local correlations. Motivated by recent advances in discriminative architectures [29, 31, 32], we design our coupling network to fuse both global context and local correlations as shown in Fig. 2: First, we project the input into a low-dimensional manifold. Next, we feed the projected tensor to a densely-connected block [29] and self-attention module [31, 33]. The densely connected block captures the local correlations [34], while the self-attention module captures the global spatial context. Outputs of these two branches are concatenated and blended through a BN-ReLU-Conv unit. As usual, the obtained output parameterizes the affine coupling transformation (15). Differences between the proposed coupling network and other network designs are detailed in related work.

It is well known that full-fledged self-attention layers have a very large computational complexity. This is especially true in the case of normalizing flows which require many coupling layers and large latent dimensionalities. We alleviate this issue by approximating the keys and queries with their low-rank approximations according to the Nystrom method [28].

## 2.3  Multi-scale architecture

We propose an image-oriented architecture which extends multi-scale Glow [19] with incremental augmentation through cross-unit coupling. Each DenseFlow block consists of several DenseFlow units and resolves a portion of the latent representation according to a decoupled normal distribution [18]. Each DenseFlow unit $f_i^{\mathrm{DF}}$ consists of $N$ glow-like modules ($m_i = m_{i,N} \circ \cdots \circ m_{i,1}$) and cross-unit coupling ($h_i$). Recall that each invertible module $m_{i,j}$ contains the affine coupling network from Fig. 2 as described Section 2.2.

The input to each DenseFlow unit is the output of the previous unit augmented with the noise and transformed in the cross-unit coupling fashion. The number of introduced noise channels is defined as the growth-rate hyperparameter. Generally, the number of invertible modules in latter DenseFlow units should increase due to enlarged latent representation. We stack $M$ DenseFlow units to form a DenseFlow block. The last invertible unit in the block does not have the corresponding cross-unit coupling. We stack multiple DenseFlow blocks to form a normalizing flow with large capacity. We decrease the spatial resolution and compress the latent representation by introducing a squeeze-and-drop modules [18] between each two blocks. A squeeze-and-drop module applies space-to-channel reshaping and resolves half of the dimensions according to the prior distribution. We denote the developed architecture as $DenseFlow\text{-}L\text{-}k$, where $L$ is the total number of invertible modules while $k$ denotes the growth rate. The developed architecture uses two independent levels of skip connections. The first level (intra-module) is formed of skip connections inside every coupling network. The second level (cross-unit) connects DenseFlow units at the top level of the architecture.

Fig. 3 shows the final architecture of the proposed model. Gray squares represent DenseFlow units. Cross-unit coupling is represented with blue dots and dashed skip connections. Finally, squeeze-and-drop operations between successive DenseFlow blocks are represented by dotted squares. The proposed DenseFlow design applies invertible but less powerful transformations (e.g. convolution $1 \times 1$) on tensors of larger dimensionality. On the other hand, powerful non-invertible transformations

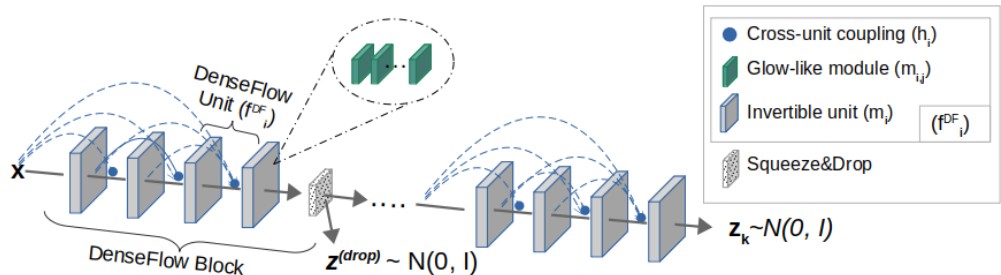

Figure 3: The proposed DenseFlow architecture. DenseFlow blocks consist of DenseFlow units ($f_i^{\mathrm{DF}}$) and a Squeeze-and-Drop module [18]. DenseFlow units are densely connected through cross-unit coupling ($h_i$). Each DenseFlow unit includes multiple invertible modules ($m_{i,j}$) from Fig. 2.

such as coupling networks perform most of their operations on lower-dimensional tensors. This leads to resource-efficient training and inference.

## 3    Experiments

Our experiments compare the proposed DenseFlow architecture with the state of the art. Quantitative experiments measure the accuracy of density estimation and quality of generated samples, analyze the computational complexity of model training, as well as ablate the proposed contributions. Qualitative experiments present generated samples.

### 3.1    Density estimation

We study the accuracy of density estimation on CIFAR-10 [35], ImageNet [36] resized to $32 \times 32$ and $64 \times 64$ pixels and CelebA [37]. Tab. 1 compares generative performance of various contemporary models. Models are grouped into four categories based on factorization of the probability density. Among these, autoregressive models have been achieving the best performance. Image Transformer [38] has been the best on ImageNet32, while Routing transformer [39] has been the best on ImageNet64. The fifth category contains hybrid architectures which combine multiple approaches into a single model. Hybrid models have succeeded to outperform many factorization-specific architectures.

The bottom row of the table presents the proposed DenseFlow architecture. We use the same DenseFlow-74-10 model in all experiments except ablations in order to illustrate the general applicability of our concepts. The first block of DenseFlow-74-10 uses 6 units with 5 glow-like modules in each DenseFlow unit, the second block uses 4 units with 6 modules, while the third block uses a single unit with 20 modules. We use the growth rate of 10 in all units. Each intra-module coupling starts with a projection to 48 channels. Subsequently, it includes a dense block with 7 densely connected layers, and the Nyström self-attention module with a single head. Since the natural images are discretized, we apply variational dequantization [27] to obtain continuous data which is suitable for normalizing flows.

On CIFAR-10, DenseFlow reaches the best recognition performance among normalizing flows, which equals to 2.98 bpd. Models trained on ImageNet32 and ImageNet64 achieve state-of-the-art density estimation corresponding to 3.63 and 3.35 bpd respectively. The obtained recognition performance is significantly better than the previous state of the art (3.77 and 3.43 bpd). Finally, our model achieves competetive results on the CelebA dataset, which corresponds to 1.99 bpd. The likelihood is computed using 1000 MC samples for CIFAR-10 and 200 samples for CelebA and ImageNet. The reported results are averaged over three runs with different random seeds. One MC sample is enough for accurate log-likelihood estimation since the per-example standard deviation is already about 0.01 bpd and a validation dataset size $N$ additionally divides it by $\sqrt{N}$. The reported results are averaged over seven runs with different random seeds. Training details are available in Appendix C.

### 3.2    Computational complexity

Deep generative models require an extraordinary amount of time and computation to reach state-of-the-art performance. Moreover, contemporary architectures have scaling issues. For example, VFlow [24] requires 16 GPUs and two months to be trained on the ImageNet32 dataset, while the NVAE [56] requires 24 GPUs and about 3 days. This limits downstream applications of developed models and slows down the rate of innovation in the field. In contrast, the proposed DenseFlow design places a great emphasis on the efficiency and scalability.

Tab. 2 compares the time and memory consumption of the proposed model with respect to competing architectures. We compare our model with VFlow [24] and NVAE [56] due to similar generative performance on CIFAR-10 and CelebA, respectively. We note that RTX 3090 and Tesla V100 deliver similar performance, while RTX2080Ti has a slightly lower performance compared to the previous two. However, since we model relatively small images, GPU utilization is limited by I/O performance. In our experiments, training the model for one epoch on any of the aforementioned GPUs had similar duration. Therefore, we can still make a fair comparison. Please note that we are unable to include approaches based on transformers [38, 39, 58] since they do not report the computational effort for model training.

Table 1: Likelihood evaluation (in bits/dim) on standard datasets.

| | Method | CIFAR-10 32x32 | ImageNet 32x32 | CelebA 64x64 | ImageNet 64x64 |
|---|---|---|---|---|---|
| Variational Autoencoders | Conv Draw [40] | 3.58 | 4.40 | - | 4.10 |
| | DVAE++ [41] | 3.38 | - | - | - |
| | IAF-VAE [42] | 3.11 | - | - | - |
| | BIVA [43] | 3.08 | 3.96 | 2.48 | - |
| | CR-NVAE [44] | **2.51** | - | **1.86** | - |
| Diffusion models | DDPM [13] | 3.70 | - | - | - |
| | UDM (RVE) + ST [45] | 3.04 | - | 1.93 | - |
| | Imp. DDPM [46] | 2.94 | - | - | 3.53 |
| | VDM [47] | 2.65 | 3.72 | - | 3.40 |
| Autoregressive Models | Gated PixelCNN [48] | 3.03 | 3.83 | - | 3.57 |
| | PixelRNN [7] | 3.00 | 3.86 | - | 3.63 |
| | PixelCNN++ [11] | 2.92 | - | - | - |
| | Image Transformer [38] | 2.90 | 3.77 | 2.61 | - |
| | PixelSNAIL [49] | 2.85 | 3.80 | - | - |
| | SPN [50] | - | 3.85 | - | 3.53 |
| | Routing transformer [39] | 2.95 | - | - | 3.43 |
| Normalizing Flows | Real NVP [18] | 3.49 | 4.28 | 3.02 | 3.98 |
| | GLOW [19] | 3.35 | 4.09 | - | 3.81 |
| | Wavelet Flow [51] | - | 4.08 | - | 3.78 |
| | Residual Flow [52] | 3.28 | 4.01 | - | 3.78 |
| | i-DenseNet [53] | 3.25 | 3.98 | - | - |
| | Flow++ [27] | 3.08 | 3.86 | - | 3.69 |
| | ANF [26] | 3.05 | 3.92 | - | 3.66 |
| | VFlow [24] | 2.98 | 3.83 | - | 3.66 |
| Hybrid Architectures | mAR-SCF [54] | 3.22 | 3.99 | - | 3.80 |
| | MaCow [55] | 3.16 | - | - | 3.69 |
| | SurVAE Flow [34] | 3.08 | 4.00 | - | 3.70 |
| | NVAE [56] | 2.91 | 3.92 | 2.03 | - |
| | PixelVAE++ [57] | 2.90 | - | - | - |
| | $\delta$-VAE [58] | 2.83 | 3.77 | - | - |
| | DenseFlow-74-10 (ours) | 2.98 | **3.63** | 1.99 | **3.35** |

Table 2: Comparative analysis of the computational budget for training contemporary methods. DenseFlow decreases the training complexity by an order of magnitude.

| Dataset | Model | Params | GPU type | GPUs | Duration (h) | BPD |
|---|---|---|---|---|---|---|
| CIFAR-10 | VFlow [24] | 38M | RTX 2080Ti | 16 | ∼500 | 2.98 |
| | NVAE [56] | 257M | Tesla V100 | 8 | 55 | 2.91 |
| | DenseFlow-74-10 | 130M | RTX 3090 | 1 | 250 | 2.98 |
| ImageNet32 | VFlow [24] | 38M | Tesla V100 | 16 | ∼1440 | 3.83 |
| | NVAE [56] | - | Tesla V100 | 24 | 70 | 3.92 |
| | DenseFlow-74-10 | 130M | Tesla V100 | 1 | 310 | 3.63 |
| CelebA | VFlow [24] | - | n/a | n/a | n/a | - |
| | NVAE [56] | 153M | Tesla V100 | 8 | 92 | 2.03 |
| | DenseFlow-74-10 | 130M | Tesla V100 | 1 | 224 | 1.99 |

## 3.3 Image generation

Normalizing flows can efficiently generate samples. The generation is performed in two steps. We first sample from the latent distribution and then transform the obtained latent tensor through the inverse mapping. Fig. 4 shows unconditionally generated images with the model trained on ImageNet64. Fig. 5 shows generated images using the model trained on CelebA. In this case, we modify the latent distribution by temperature scaling with factor 0.8 [38, 19, 56]. Generated images show diverse hairstyles, skin tones and backgrounds. More generated samples can be found in Appendix G. The

developed DenseFlow-74-10 model generates minibatch of 128 CIFAR-10 samples for 0.96 sec. The result is averaged over 10 runs on RTX 3090.

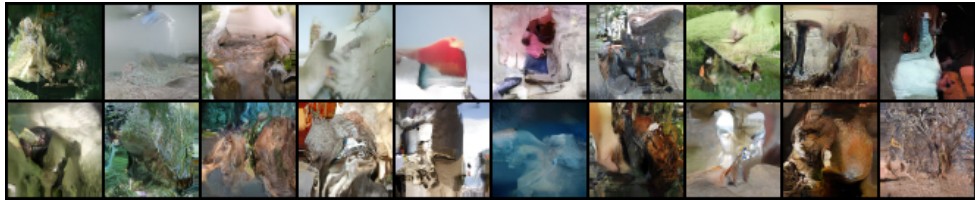

Figure 4: Samples from DenseFlow-74-10 trained on ImageNet $64 \times 64$.

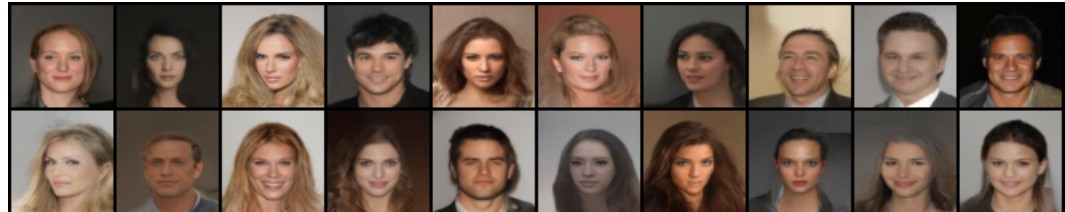

Figure 5: Samples from DenseFlow-74-10 trained on CelebA.

## 3.4 Visual quality

The ability to generate high fidelity samples is crucial for real-world applications of generative models. We measure the quality of generated samples using the FID score [59]. The FID score requires a large corpus of generated samples in order to provide an unbiased estimate. Hence, we generate 50k samples for CIFAR-10, and CelebA, and 200k samples for ImageNet. The samples are generated using the model described in Sec. 3.1. The generated ImageNet32 samples achieve a FID score of 38.8, the CelebA samples achieve 17.1 and CIFAR-10 samples achieve 34.9 when compared to the corresponding training dataset. When compared with corresponding validation datasets, we achieve 37.1 on CIFAR10 and 38.5 on ImageNet32.

Tab. 3 shows a comparison with FID scores of other generative models. Our model outperforms contemporary autoregressive models [7, 60] and the majority of normalizing flows [61, 52, 19]. Our FID score is comparable with the first generation of GANs. Similar to other NF models, the achieved FID score is still an order of magnitude higher than current state of the art [62]. The results for PixelCNN, DCGAN, and WGAN-GP are taken from [60].

## 3.5 Ablations

Tab. 4 explores the contributions of incremental augmentation and dense connectivity in cross-unit and intra-module coupling transforms. We decompose cross-unit coupling into incremental augmentation of the flow dimensionality (column 1) and affine noise transformation (column 2). Column 3 ablates the proposed intra-module coupling network based on fusion of fast self-attention and a densely connected convolutional block with the original Glow coupling [19].

The bottom row of the table corresponds to a DenseFlow-45-6 model. The first DenseFlow block has 5 DenseFlow units with 3 invertible modules per unit. The second DenseFlow block has 3 units with 5 modules, while the final block has 15 modules in a single unit. We use the growth rate of 6. The top row of the table corresponds to the standard normalized flow [18, 19] with three blocks and 15 modules per block. Consequently, all models have the same number of invertible glow-like modules. All models are trained on CIFAR-10 for 300 epochs and then fine-tuned for 10 epochs. We use the same training hyperparameters for all models. The proposed cross-unit coupling improves the density estimation from 3.42 bpd (row 1) to 3.37 bpd (row 3) starting from a model with the standard glow modules. When a model is equipped with our intra-module coupling, cross-unit coupling leads to improvement from 3.14 bpd (row 4) to 3.07 bpd (row 6). Hence, the proposed cross-unit coupling

Table 3: Evaluation of FID score on CIFAR-10.

|  | Model | FID ↓ |
|---|---|---|
| Autoregressive Models | PixelCNN [7, 60] | 65.93 |
|  | PixelIQN [60] | 49.46 |
| Normalizing Flows | i-ResNet [61] | 65.01 |
|  | Glow [19] | 46.90 |
|  | Residual flow [52] | 46.37 |
|  | ANF [26] | 30.60 |
| GANs | DCGAN [15, 60] | 37.11 |
|  | WGAN-GP [63, 60] | 36.40 |
|  | DA-StyleGAN V2 [62] | 5.79 |
| Diffusion models | VDM [47] | 4.00 |
|  | DDPM [13] | 3.17 |
|  | UDM (RVE) + ST [45] | 2.33 |
| Hybrid Architectures | SurVAE-flow [34] | 49.03 |
|  | mAR-SCF [54] | 33.06 |
|  | VAEBM [64] | 12.19 |
|  | DenseFlow-74-10 (ours) | 34.90 |

improves the density estimation in all experiments. Both components of cross-unit coupling are important. Models with preconditioned noise outperform models with simple white noise (row 2 vs row 3, and row 5 vs row 6). A comparison of rows 1-3 with rows 4-6 reveals that the proposed intra-module coupling network also yields significant improvements. We have performed two further ablation experiments with the same model. Densely connected cross-coupling contributes 0.01 bpd in comparison to preconditioning noise with respect to the previous representation only. Self-attention module contributes 0.01 bpd with respect to the model with only DenseBlock coupling on ImageNet $32 \times 32$.

Table 4: Ablations on the CIFAR-10 dataset with DenseFlow-45-6.

| # | Latent variable augmentation | Pre-conditioned noise | Intra-module coupling with two-way fusion | BPD |
|---|---|---|---|---|
| 1 | ✗ | ✗ | ✗ | 3.42 |
| 2 | ✓ | ✗ | ✗ | 3.40 |
| 3 | ✓ | ✓ | ✗ | 3.37 |
| 4 | ✗ | ✗ | ✓ | 3.14 |
| 5 | ✓ | ✗ | ✓ | 3.08 |
| 6 | ✓ | ✓ | ✓ | 3.07 |

# 4 Related work

VFlow [24] increases the dimensionality of a normalizing flow by concatenating input with a random variable drawn from $p(e|x)$. The resulting optimization maximizes the lower bound $\mathbb{E}_{e \sim p^*(e|x)}[\ln p(x, e) - \ln p^*(e|x)]$, where each term is implemented by a separate normalizing flow. Similarly, ANF [26] draws a connection between maximizing the joint density $p(x, e)$ and lower-bound optimization [4]. Both approaches augment only the input variable $x$ while we augment latent representations many times throughout our models.

Surjective flows [34] decrease the computational complexity of the flow by reducing the dimensionality of deep layers. However, this also reduces the generative capacity. Our approach achieves a better generative performance under affordable computational budget due to gradual increase of the latent dimensionality and efficient coupling.

Invertible DenseNets [65, 53] apply skip connections within invertible residual blocks [61, 52]. However, this approach lacks a closed-form inverse, and therefore can generate data only through slow iterative algorithms. Our approach leverages skip connections both in cross-unit and intra-module couplings, and supports fast analytical inverse by construction.

Models with an analytical inverse allocate most of their capacity to coupling networks [17]. Early coupling networks were implemented as residual blocks [18]. Recent work [27] increases the coupling capacity by stacking convolutional and multihead self-attention layers into a gated residual [66, 49]. However, heavy usage of self-attention radically increases the computational complexity. Contrary to stacking convolutional and self-attention layers in alternating fashion, the design of our network uses two parallel modules. Outputs of the these two modules are fused into a single output. SurVAE [34] expresses the coupling network as a densely connected block [29] with residual connection. In comparison with [34], our intra-module coupling omits residual connectivity, decreases the number of densely connected layers and introduces a parallel branch with Nyström self-attention. Thus, our intra-module coupling fuses local cues with the global context.

Normalizing flow capacity can be further increased by adding complexity to the latent prior $p(\boldsymbol{z})$. Autoregressive prior [54] may deliver better density estimation and improved visual quality of the generated samples. However, the computational cost of sample generation grows linearly with spatial dimensionality. Joining this approach with the proposed incremental latent variable augmentation could be a suitable direction for future work.

## 5   Conclusion

Normalizing flows allow principled recovery of the likelihood by evaluating factorized latent activations. However, their efficiency is hampered by the bijectivity constraint since it determines the model width. We propose to address this issue by incremental augmentation of intermediate latent representations. The introduced noise is preconditioned with respect to preceding representations throughout cross-unit affine coupling. We also propose an improved design of intra-module coupling transformations within glow-like invertible modules. We express these transformations as a fusion of local correlations and the global context captured by self-attention. The resulting DenseFlow architecture sets the new state-of-the-art in likelihood evaluation on ImageNet while requiring a relatively small computational budget. Our results imply that the expressiveness of a NF does not only depend on latent dimensionality but also on its distribution across the model depth. Moreover, expressiveness of a NF can be further improved by conditioning the introduced noise with the proposed densely connected cross-unit coupling.

## 6   Broader impact

This paper introduces a new generative model called DenseFlow, which can be trained to achieve state-of-the-art density evaluation under moderate computational budget. Fast convergence and modest memory footprint lead to relatively small environmental impact of training and favor applicability to many downstream tasks. Technical contributions of this paper do not raise any particular ethical challenges. However, image generation has known issues related to bias and fairness [67]. In particular, samples generated by our method will reflect any kind of bias from the training dataset.

## Acknowledgements

This work has been supported by Croatian Science Foundation, grant IP-2020-02-5851 ADEPT. The first two authors have been employed on research projects KK.01.2.1.02.0119 DATACROSS and KK.01.2.1.02.0119 A-Unit funded by European Regional Development Fund and Gideon Brothers ltd. This work has also been supported by VSITE - College for Information Technologies who provided access to 2 GPU Tesla-V100 32GB. We thank Marin Oršić, Julije Ožegović, Josip Šarić as well as Jakob Verbeek for insightful discussions during early stages of this work.

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
