## A  Limitations

The proposed DenseFlow model is based on the extended NF framework. However, since it uses Monte Carlo sampling by construction to estimate the likelihood, its characteristics slightly deviate from standard NF models.

Aggressive application of the cross-unit coupling can lead to overgrowth of the latent tensor. Therefore, the developed architecture can become computationally intractable. This problem can be alleviated by using an appropriate growth rate and careful application of the cross-unit coupling step.

Replacing standard glow modules [19] with the proposed glow-like modules which fuse local correlations and global context increases the expressiveness of the resulting normalizing flow at cost of an increased number of trainable parameters.

## B  Environmental impact

Our DenseFlow model achieves competitive results with significantly less computation. Our experiments were conducted using private infrastructure powered by public grid, which has a carbon efficiency of 0.329 $kgCO_2eq$/kWh. A cumulative of 1120 hours of computation was performed for the main experiments. According to [68], total emissions are estimated to be 110.54 $kgCO_2eq$.

## C  Training details

We train the proposed DenseFlow-74-10 architecture on ImageNet32 for 20 epoch using Adamax optimizer with learning rate set to $10^{-3}$ and batch size 64. We augment the training data by applying random horizontal flip with the probability of 0.5. We apply linear warm-up of the learning rate in the first 5000 epochs. During training, learning rate is exponentially decayed by a factor of 0.95 after every epoch. The model is fine-tuned using a learning rate of $2 \cdot 10^{-5}$ for 2 epochs. Similarly, the model is trained for 10 epoch on ImageNet64, 50 epochs on CelebA and 580 epochs on CIFAR-10. In the case of CIFAR-10, we decay the learning rate by a factor of 0.9975. The model is fine-tuned for 1 epoch on ImageNet64, 5 epochs on CelebA and 70 epochs on CIFAR-10. We use batch size of 64 for CIFAR-10 and 32 for CelebA and ImageNet64. Other hyperparameters are the same as in ImageNet32 training.

## D  Proofs

### D.1  Proof of Equation (5)

We denote target distributions by $p^*$ and learned distributions by $p$. Let $\boldsymbol{z}_i$ denote the input, which we consider to be distributed according to $p^*(\boldsymbol{z}_i)$. Let $\boldsymbol{e}_i$ be noise independent of $\boldsymbol{z}_i$ with a known distribution $p^*(\boldsymbol{e}_i)$. Let $p(\boldsymbol{h}_i)$ be a Gaussian distribution and $f$ a function representing a normalizing flow: $\boldsymbol{h}_i = f(\boldsymbol{z}_i, \boldsymbol{e}_i)$. The normalizing flow distribution

$$p(\boldsymbol{z}_i, \boldsymbol{e}_i) = p(\boldsymbol{h}_i) \left| \det \frac{\partial \boldsymbol{h}_i}{\partial(\boldsymbol{z}_i, \boldsymbol{e}_i)} \right| \tag{16}$$

approximates the true distribution $p^*(\boldsymbol{z}_i, \boldsymbol{e}_i)$, which is factorized as the product of $p^*(\boldsymbol{z}_i)$ and $p^*(\boldsymbol{e}_i)$. We do not have a guarantee that $p(\boldsymbol{z}_i) = p(\boldsymbol{z}_i, \boldsymbol{e}_i)/p^*(\boldsymbol{e}_i)$.

To get the density $p(\boldsymbol{z}_i)$, we have to marginalize $p(\boldsymbol{z}_i, \boldsymbol{e}_i)$:

$$p(\boldsymbol{z}_i) = \int p(\boldsymbol{z}_i, \boldsymbol{e}_i) \, d\boldsymbol{e}_i \,. \tag{17}$$

We can efficiently estimate the integral using importance sampling:

$$p(\boldsymbol{z}_i) = \int \frac{p(\boldsymbol{z}_i, \boldsymbol{e}_i)}{p^*(\boldsymbol{e}_i)} p^*(\boldsymbol{e}_i) \, d\boldsymbol{e}_i \tag{18}$$

$$= \mathbb{E}_{\boldsymbol{e}_i \sim p^*(\boldsymbol{e}_i)} \left[ \frac{p(\boldsymbol{z}_i, \boldsymbol{e}_i)}{p^*(\boldsymbol{e}_i)} \right] . \tag{19}$$

Log-likelihood can be computed as:

$$\ln p(\boldsymbol{z}_i) = \ln \mathbb{E}_{\boldsymbol{e}_i \sim p^*(\boldsymbol{e}_i)} \left[ \frac{p(\boldsymbol{z}_i, \boldsymbol{e}_i)}{p^*(\boldsymbol{e}_i)} \right] . \tag{20}$$

By applying Jensen's inequality, we obtain a lower bound on the log-likelihood,

$$\ln p(\boldsymbol{z}_i) \geq \mathbb{E}_{\boldsymbol{e}_i \sim p^*(\boldsymbol{e}_i)} \left[ \ln p(\boldsymbol{z}_i, \boldsymbol{e}_i) - \ln p^*(\boldsymbol{e}_i) \right], \tag{21}$$

which corresponds to Eq. (5).

# E    Evidence against training set memorization

It can be argued that increasing the dimensionality of the latent representation can lead to training set memorization. We show that this is not the case when using DenseFlow. Following [56], we first generate faces using the model trained on CelebA, and then compute the $L_2$ distance between the generated and the training images. The $L_2$ distance is computed over $42 \times 42$ center crop, so it captures only face pixels. Fig. 6 shows generated images in the first column, followed by the five most similar training images.

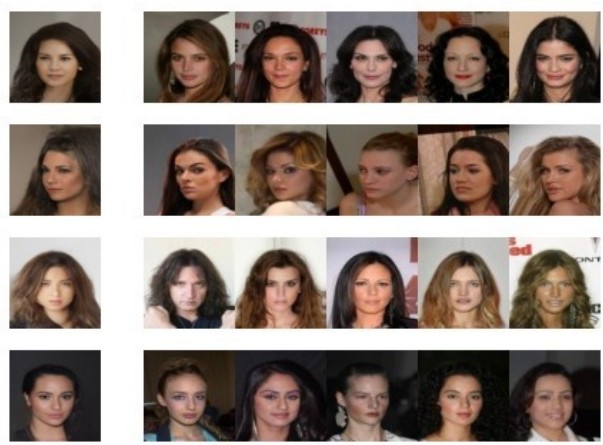

Figure 6: The generated faces and the five most similar samples from training set.

# F    Implementation

Significant part of our experimental implementation benefited from [34] whose code was publicly released under the MIT license. Our code can be found in supplementary material together with the link to model parameters.

# G    More samples

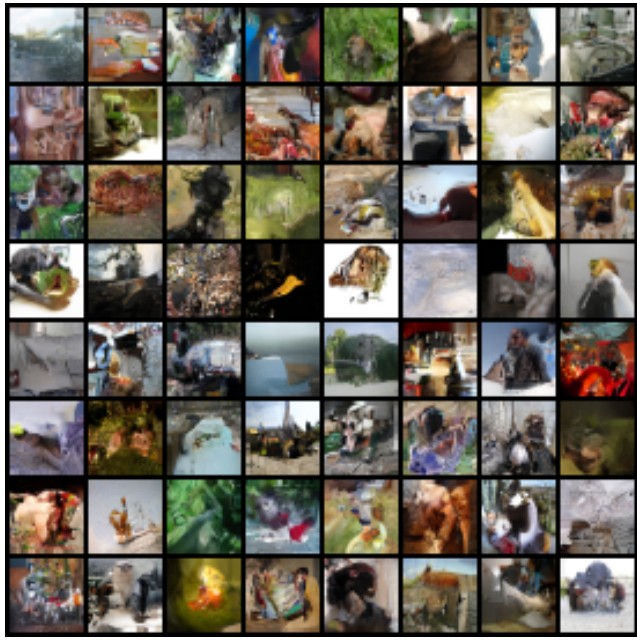

Figure 7: ImageNet $32 \times 32$ samples.

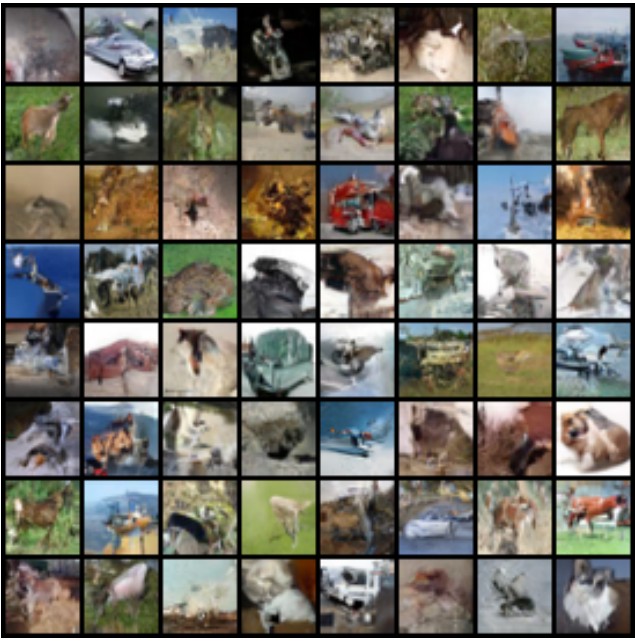

Figure 8: CIFAR-10 samples.

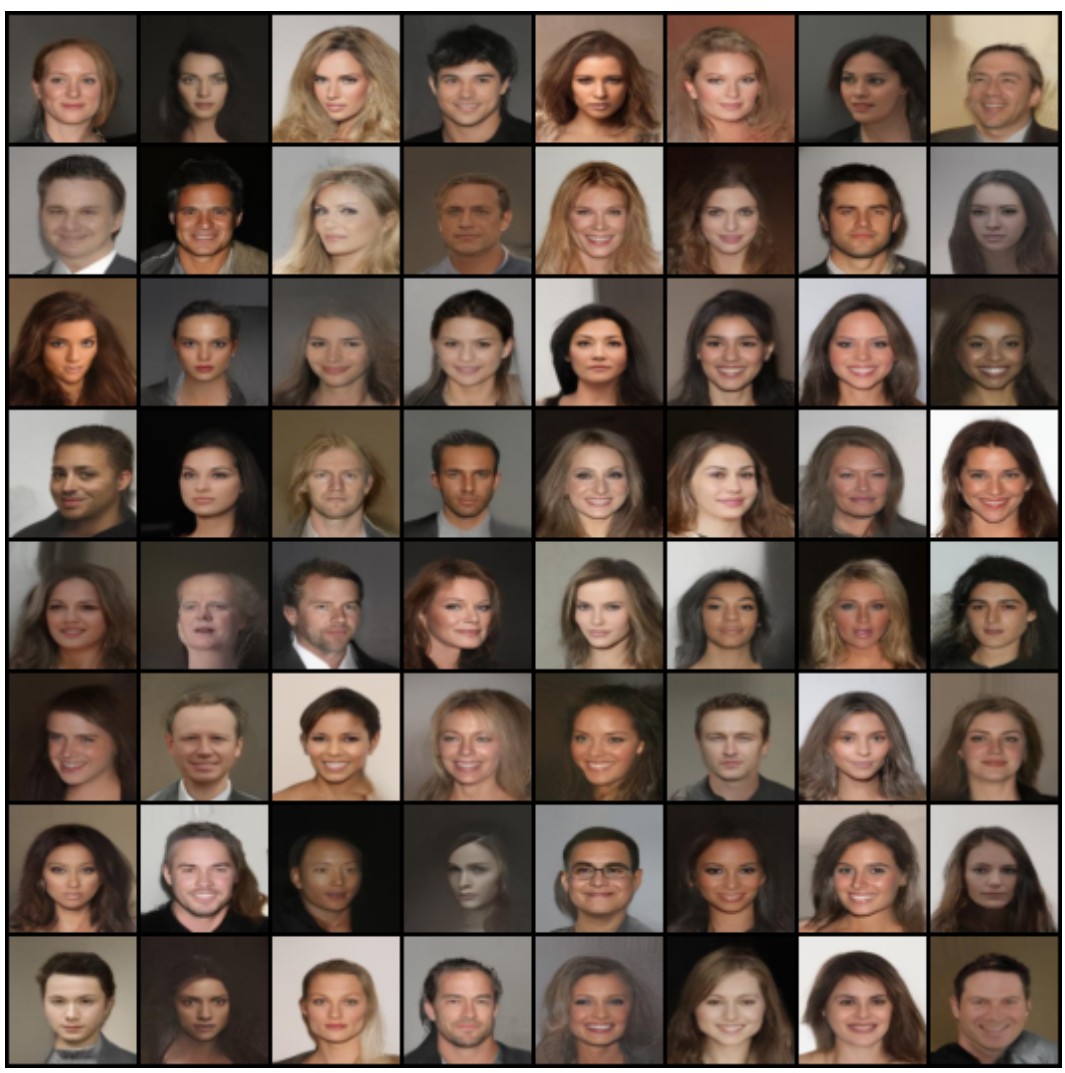

Figure 9: CelebA samples.

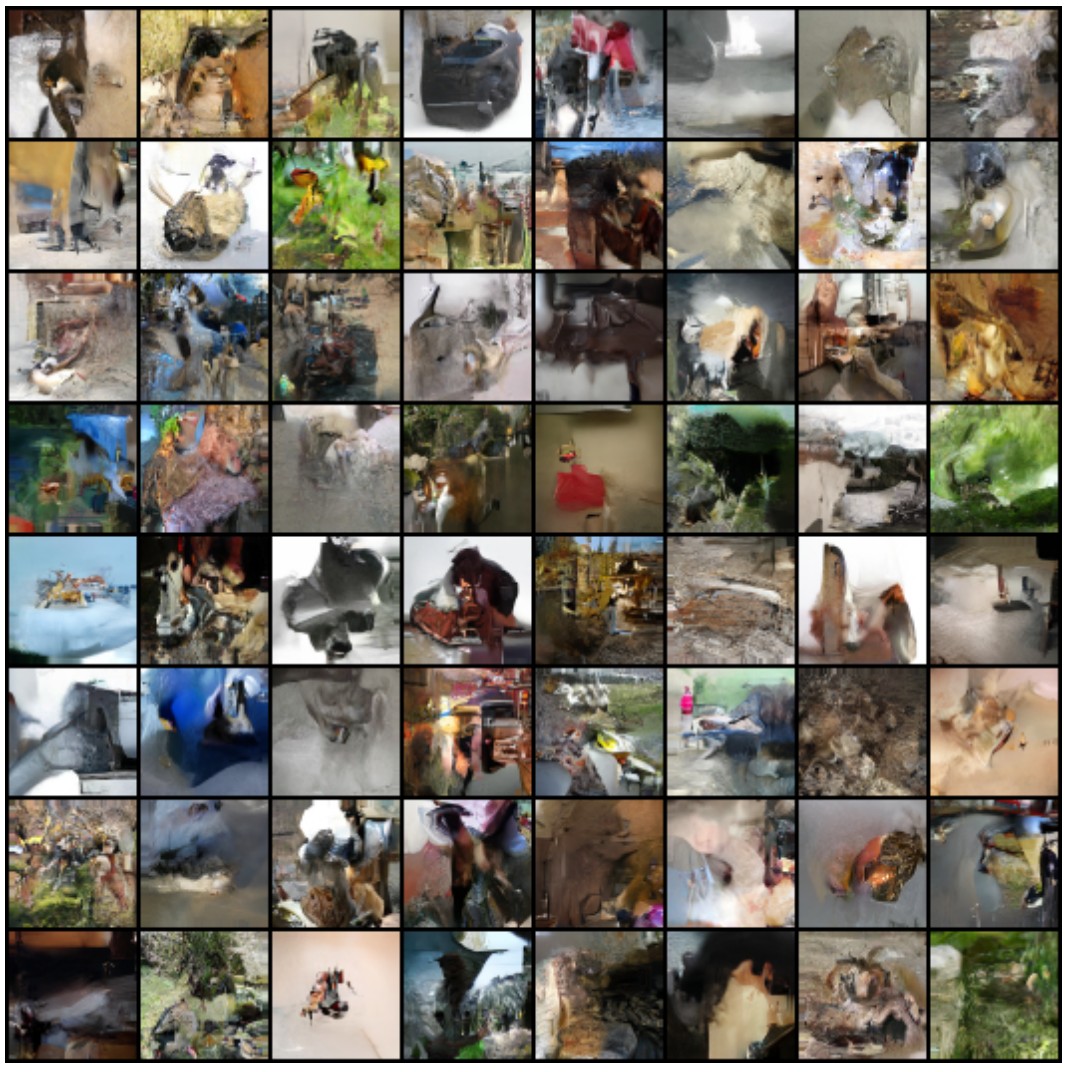

Figure 10: ImageNet $64 \times 64$ samples.