# OpenReview forum: "Densely connected normalizing flows"
_NeurIPS.cc/2021/Conference — NeurIPS 2021 Poster_

### Official Review · Reviewer_RyGz · 2021-07-12

**Rating:** 7
**Confidence:** 4

**Summary:**

The paper proposes to use incremental augmentation, gradually increasing dimensionality towards the latent space, in normalizing flows. The augmentation distribution at each level is conditioned on previous feature maps. This is referred to as "cross-unit coupling". Additionally, the authors introduce a new neural architecture for use in the usual coupling layers, referred to as "intra-unit coupling".
The result is an efficiently trained flow with state-of-the-art likelihoods.

**Limitations And Societal Impact:**

Limititations are adequately discussed in App. A and environmental impact in App. B.
As for negative societal imapct, the authors mention that their research could be used for malicious purposes, they do not discuss any particular examples.

**Main Review:**

The paper extends the augmentation in [1,2] to multiple layers where the augmentation distributions are conditioned on previous feature maps in a similar fashion to DenseNets. Furthermore, a novel neural architecture is proposed for the coupling layers.

The proposed improvements are straightforward extensions of existing methodology, but with surprisingly good performance. The proposed methodology is clearly laid out, first introducing the "cross-unit coupling", before introducing the image-oriented neural architecture with two-way fusion using both convolutions and self-attention.
Experiments are quite extensive: Comparisons on likelihood performance on 4 image datasets, as well as comparisons of computational complexity and sample quality. Finally, there is also an ablation study on the different model component's effect on likelihood.

The main strong point of the paper is the state-of-the-art likelihood performance achieved using modest computational resources compared to earlier work. Given this, I am recommending acceptance. However, the paper does have some rough edges and points that could be improved. I outline a few suggestions for this below.

Some suggestions for fixing/improving the paper:
- Line 14: "One of the uttermost goals [...]": This statement is maybe a bit strong and could be rephrased.
- Line 27: Although the connections you point out are correct, I'm not sure I'd call diffusion models as VAEs since there is no "autoencoding" going on.
- Line 29: "[...] GANs ignore the factorization of the likelihood": What is meant by this? I think the description of GANs could be improved. The inability to evaluate the likelihood comes mainly from the inability to "invert" the generation process in any meaningful way.
- Eq. 11: $e_i \rightarrow e_1$ and $\log p(e_1) \rightarrow \log p^*(e_1)$.
- Line 108: "[...] a few hundred of samples during evaluation to reduce the variance of the likelihood": From the code, it seems the importance weighted bound was used which also provides a tighter bound on the likelihood, not just lower variance.
- Table 2: Do you also have access to the number of parameters? If so, this would improve the table.
- Line 241: While one could argue about which approach is best, it is my understanding that common practice is to compute FID by comparing the training data distribution.
- Line 263: bdp -> bpd
- Line 276: $\mathbb{E}_{\boldsymbol{e} \sim p(\boldsymbol{e}|\boldsymbol{x})} \left[ \log p(\boldsymbol{e}, \boldsymbol{x}) - \log p(\boldsymbol{e}|\boldsymbol{x}) \right] = \log p(\boldsymbol{x})$ and so is not a lower bound.
  - I believe you want to write $\mathbb{E}_{\boldsymbol{e} \sim p^*(\boldsymbol{e}|\boldsymbol{x})} \left[ \log p(\boldsymbol{e}, \boldsymbol{x}) - \log p^*(\boldsymbol{e}|\boldsymbol{x}) \right]$



[1] Huang et al. 2020, Augmented Normalizing Flows: Bridging the Gap Between Generative Flows and Latent Variable Models
[2] Chen et al. 2020, VFlow: More Expressive Generative Flows with Variational Data Augmentation

**Time Spent Reviewing:**

5

---

> ### Author Response · Authors · 2021-08-10
> **Authors response to Reviewer RyGz**
>
> We would like to thank the reviewer for constructive feedback.
>
> > The proposed improvements are straightforward extensions of existing methodology
>
> We show that the gradual increase in latent dimensionality along the depth is more advantageous than feeding equally sized tensors throughout the whole model. Initial layers encode simpler data patterns into the latent tensor, while the latter layers encode more complex patterns. An additional expansion of the latent tensor in the latter stages enables the model to capture an additional amount of complex patterns, which leads to better results.
>
> > Line 14: “One of the uttermost goals […]”: This statement is maybe a bit strong and could be rephrased.
>
> We agree with the reviewer that this phrase is not very well written. We will rephrase it in the next version of the manuscript.
>
> > Line 27: Although the connections you point out are correct, I’m not sure I’d call diffusion models as VAEs since there is no “autoencoding” going on.
>
>  This statement is motivated by [13], where authors draw connections between DDPM and VAEs. We agree that these two models should be described separately and will revise them in the next version of the manuscript.
>
> > Line 29: “[…] GANs ignore the factorization of the likelihood”: What is meant by this? I think the description of GANs could be improved. The inability to evaluate the likelihood comes mainly from the inability to “invert” the generation process in any meaningful way.
>
> We meant to say that GANs do not attempt to model individual factors of variation through a factorized formulation of p(x). We will clarify this in the next revision.
>
> > Eq. 11: ei→e1 and log⁡p(e1)→log⁡p∗(e1).
> >  Line 276: Ee∼p(e|x)[log⁡p(e,x)−log⁡p(e|x)]=log⁡p(x) and so is not a lower bound.
>     I believe you want to write Ee∼p∗(e|x)[log⁡p(e,x)−log⁡p∗(e|x)]
> >  Line 263: bdp -> bpd
>
> We thank for the correction! We shall improve the notation consistency in the next revision.
>
> > Line 108: “[…] a few hundred of samples during evaluation to reduce the variance of the likelihood”: From the code, it seems the importance weighted bound was used which also provides a tighter bound on the likelihood, not just lower variance.
>
> You are right. We will point it out in the manuscript.
>
> > Table 2: Do you also have access to the number of parameters? If so, this would improve the table.
>
> Some of the papers from Table 2 reveal the number of parameters. We shall present the number of parameters as an additional column in Table 2. DenseFlow-74-10 has 130M parameters.
>
> > Line 241: While one could argue about which approach is best, it is my understanding that common practice is to compute FID by comparing the training data distribution.
>
> We achieve somewhat better FID performance when evaluating on the training data (e.g. 37 for CIFAR10). We shall include training FID in Table 3.

---

> > ### Comment · Reviewer_RyGz · 2021-08-25
> > **Post-rebuttal**
> >
> > Thank you for your feedback. One of the main concerns among reviewers is novelty. I tend to agree here, but also think the performance gain itself renders the work interesting. Without naming specific papers, I also find lower degrees of novelty to not be uncommon for papers demonstrating new state-of-the-art, e.g. in the literature on VAEs and autoregressive models. I am thus still in favor of accepting, but if other reviewers and/or the area chair find the paper to be too incremental, I am also ok with this.

---

### Official Review · Reviewer_BeKy · 2021-07-17

**Rating:** 7
**Confidence:** 4

**Summary:**

This paper proposes an approach to improve the modelling capacity of normalising flow based models. The proposed approach  incrementally pads intermediate representations with noise, which is conditioned on the output of previous invertible units using affine transformations. The paper reports promising results density estimation results on ImageNet and CelebA.

**Limitations And Societal Impact:**

* The paper does not motivate the novelty in term of architecture over prior methods such as Flow++.
* The computational complexity for likelihood estimation should be discussed in more detail e.g. the trade-off of using higher number of samples vs bpd.
* The poor performance in terms of image quality must be better motivated and comparison to prior work such as [Normalizing Flows with Multi-Scale Autoregressive Priors, CVPR 2020] must be included.

**Main Review:**

The proposed approach for noise augmentation is interesting and novel. However, the main concerns are,
* Motivation -- the paper does not clearly state the advantage of their approach over VFlows. In other words, why does the proposed approach perform better than VFlows? L43-51 only establishes the advantage of augmenting extra dimensions and Fig 1 only illustrates the difference to VFlows.

* Architecture of glow like modules -- The paper proposes to include Nyström attention modules, however, Flow++ already includes multi-head attention (Section 2.2 should therefore discuss the difference to Flow++ in more detail). The main advantage of Nyström attention lies in computational complexity. However, there are no ablations to study this advantage in detail. Therefore, the architectural advantage of the proposed approach over Flow++ is unclear.

* Density estimation -- the paper does not motivate why the results on CIFAR-10 do not show any advantage over VFlow (same bpd).

* Efficiency of likelihood computation -- the paper should include a detailed analysis of speed of likelihood computation as 1000 MCMC samples are needed versus flow based models which provide likelihood estimates in a single pass.

* Sample quality -- the FID score on CIFAR-10 is competitive with other flow based approaches such as, [Normalizing Flows with Multi-Scale Autoregressive Priors, CVPR 2020] which achieves an FID of 33.6 on CIFAR-10. Moreover, there is no comparison in terms of image quality to VFlow. The paper should motivate why the proposed approach fails to improve in terms of image quality compared to prior flow based models.







**Time Spent Reviewing:**

3

---

> ### Author Response · Authors · 2021-08-10
> **Authors response to Reviewer BeKy**
>
> We would like to thank the reviewer for insightful feedback.
>
> > the paper does not clearly state the advantage of their approach over VFlows. In other words, why does the proposed approach perform better than VFlows?
>
> We show that the gradual increase in latent dimensionality along the depth is more advantageous than feeding equally sized tensor throughout the whole model. Initial layers encode simpler data patterns into the latent tensor, while the latter layers encode more complex patterns. Expansion of the latent tensor in the latter stages enables the model to capture an additional amount of complex patterns. Also, this is the first NF formulation that increases the latent representation along with the model depth. Our improvements lead to faster training, smaller memory footprint and better recognition performance.
>
> >  Section 2.2 should therefore discuss the difference to Flow++ in more detail
>
> Every coupling network of Flow++ (and VFlow) is formed as a stack of 10 blocks. Every block contains a gated self-attention layer followed by a gated convolutional layer. The gating function splits the input into two halves (h1, h2) and computes h1 * \sigm(h2) as proposed in [46,61]. Normalization layers are inserted after each attention and after each convolution. Finally, these two layers are residually connected. Contrary, we feed the same input to a dense block [29] and a self-attention block (cf. Fig.2). Outputs of these two blocks are concatenated and blended by a single convolutional unit. Since the self-attention block is memory intensive, we approximate it with the Nystrom method [28]. The proposed coupling is less memory-intensive than alternatives from all existing normaiizing flows, which is important for many downstream tasks.
>
> > However, there are no ablations to study this advantage in detail. Therefore, the architectural advantage of the proposed approach over Flow++ is unclear.
>
> The proposed model requires 3.1 GB of memory for the forward pass and backward gradient computation. In comparison, when we replace our intra-unit couplings with Flow++/VFlow couplings, the model consumes 20.7 GB of GPU memory. The measurement is obtained by torch.cuda.max_memory_allocated() on RTX3090 using a minibatch of 16 32x32 images.
> Hence, the proposed coupling leads to significant memory savings.
>
> > the paper does not motivate why the results on CIFAR-10 do not show any advantage over VFlow (same bpd).
>
> Due to significant differences in architectures (e.g. VFlow concatenates the input x with noise e sampled from p(e|x), where p(e|x) is defined using an additional NF flow) we can only provide an educated guess about causes of similar BPD.
> Still, DenseFlow compares favorably with VFlow on CIFAR-10. We achieve similar density estimation performance while using significantly less computational resources.
>
> > the FID score on CIFAR-10 is competitive with other flow based approaches such as, [Normalizing Flows with Multi-Scale Autoregressive Priors, CVPR 2020] which achieves an FID of 33.6 on CIFAR-10.
>
> This paper introduces autoregressive priors for normalizing flows, while we use simple Gaussian priors. Extending the proposed DenseFlow by employing autoregressive priors could be interesting further work. We will discuss the proposed work in future revisions of this manuscript.
> Contrary to reviewer's comment, the mAR-SCF model from [Normalizing Flows with Multi-Scale Autoregressive Priors, CVPR 2020] reports 41.0 as the FID score for CIFAR10. Did we miss something?
>
> > Moreover, there is no comparison in terms of image quality to VFlow.
>
> Unfortunately, VFlow paper does not report FID score. Due to the usage of legacy components in the official code, we couldn't compute the score by ourselves in time for this discussion. We attempted to reach out to the authors of VFlow but did not receive any feedback.
>
> > The paper should motivate why the proposed approach fails to improve in terms of image quality compared to prior flow based models.
> > The poor performance in terms of image quality must be better motivated
>
> DenseFlow outperforms the Glow with respect to FID score. However, since our model builds upon NF, we inherit NF inductive biases which result in significantly lower FID compared to modern GANs. Improving the image quality is a one of directions of future work.
>
> > The paper does not motivate the novelty in term of architecture over prior methods such as Flow++.
>
> High computational requirements of Flow++ limit the usage of the model in many downstream tasks. The main cause of high computational requirements is a coupling network architecture with multiple multi-head attention modules per block. We propose a simpler coupling with smaller memory footprint and fewer parameters.
>
> > The computational complexity for likelihood estimation should be discussed in more detail e.g. the trade-off of using higher number of samples vs bpd.
>
> > the paper should include a detailed analysis of speed of likelihood computation as 1000 MCMC samples are needed versus flow based models which provide likelihood estimates in a single pass.
>
> We have now estimated the standard deviation of estimating likelihood in one validation input. The standard deviation with 1 MC sample is about 0.01 bpd. Thus, the standard deviation when using $M$ MC samples and $N$ validation examples is about $\sqrt{MN}$ times smaller. I.e. it is insiginificant even with $M=1$ and we could just as well have used 1 instead of 1000 MC samples.

---

> > ### Comment · Reviewer_BeKy · 2021-08-26
> > **Post rebuttal**
> >
> > The rebuttal addresses most of my concerns. Although the novelty is somewhat limited, overall the paper shows strong results to merit acceptance. Note: the workshop version of [Normalizing Flows with Multi-Scale Autoregressive Priors, CVPR 2020] in INNF+ (ICML 2020) shows improved FID of 33.6 on CIFAR-10 (https://invertibleworkshop.github.io/INNF_2020/accepted_papers/pdfs/23.pdf).

---

### Official Review · Reviewer_EGVd · 2021-07-17

**Rating:** 6
**Confidence:** 4

**Summary:**

This work addresses the limitation posed by the bijectivity constraints in normalizing flows by padding  noise at each layer of the flow.
MC sampling is used to approximate the likelihood where 1 MC sample is used for training and multiple MC samples are used for likelihood evaluation. Experiments for density estimation are performed on CIFAR-10, ImageNet-32 ,CelebA -64, ImageNet-64 datasets where the method yields state of the art density estimates. The method is shown to be computationally efficient with better sample quality.

**Limitations And Societal Impact:**

The limitations and the societal impact of the work are appropriately discussed.

**Main Review:**

* The work is incremental as the idea of padding with the noise dimensions to overcome the constraints associated with the dimensionality preservation have been studied in INNs and VFlow.

* Overall the paper is well written and the experiments are detailed and easy to follow. Relevant details wrt to the experimental setup are included.  Tables 1, 2 and 4 include relevant comparisons to the prior work for density estimation, computational complexity and ablations of the proposed work respectively. The FID scores in table 3 should be expanded with comparison to other flow models that yield better sample quality e.g. [1,2].

* The work proposes an intra-coupling module in section 2.2 . The details of the same can be elaborated as the architecture and the operations performed are difficult to follow. The self attention module is similar to that of Flow++. A discussion regarding the differences of the proposed approach with that of the existing self-attention module (of Flow++) would be helpful.

* Choice of the number of MC samples for different datasets in line 211 can be expanded and the variance thereof discussed. How is the training effected due to 1 MC sample?

* The related work section seems to be minimal and should be expanded to include relevant literature on normalizing flows and explicit generative models in general. The related work of [1,2,3] can be referred for details.


[1]. Bhattacharyya et al. Normalizing Flows with Multi-Scale Autoregressive Priors. CVPR 2020.
[2]. Ma et al. MaCaw: masked convolutional generative flow. NeurIPS 2019.
[3]. Yu et al. Wavelet Flow: Fast Training of High Resolution Normalizing Flows. NeurIPS 2020.

**Time Spent Reviewing:**

16

---

> ### Author Response · Authors · 2021-08-10
> **Authors response to Reviewer EGVd**
>
> We would like to thank the reviewer for helpful feedback.
>
> > The work is incremental as the idea of padding with the noise dimensions to overcome the constraints associated with the dimensionality preservation have been studied in INNs and VFlow.
>
> We agree with the reviewer that augmenting the input with noise channels has been proposed in earlier work, as discussed in section 4. However, our manuscript presents two additional insights which make a considerable influence to the NF performance. Our first insight is that expressiveness of a NF does not only depend on network width but also on its distribution across the depth. Our second insight is that the expressiveness can also be improved by conditioning the noise with densely connected cross-unit coupling.
> A combination of these two insights and the proposed memory-efficient intra-unit couplings substantially reduces computational requirements for training normalizing flows. This result could promote usage of deep generative models in new downstream tasks.
>
> > The FID scores in table 3 should be expanded with comparison to other flow models that yield better sample quality e.g. [1,2].
>
> We will expand Table 3 with additional methods including [1]. The method [2] does not report FID score. Still, the density estimation performance of [2] can be found in Table 1.
>
> > The work proposes an intra-coupling module in section 2.2 . The details of the same can be elaborated as the architecture and the operations performed are difficult to follow. The self attention module is similar to that of Flow++. A discussion regarding the differences of the proposed approach with that of the existing self-attention module (of Flow++) would be helpful.
>
> Every coupling network of Flow++ (and VFlow) is formed as a stack of 10 blocks. Every block contains a gated self-attention layer followed by a gated convolutional layer. The gating function splits the input into two halves (h1, h2) and computes h1 * \sigm(h2) as proposed in [46,61]. Normalization layers are inserted after each attention and after each convolution. Finally, these two layers are residually connected. Contrary, we feed the same input to a dense block [29] and a self-attention block (cf. Fig.2). Outputs of these two blocks are concatenated and blended by a single convolutional unit. Since the self-attention block is memory intensive, we approximate it with the Nystrom method [28]. The proposed coupling is less memory-intensive than alternatives from all existing normaiizing flows, which is important for many downstream tasks.
>
> > Choice of the number of MC samples for different datasets in line 211 can be expanded and the variance thereof discussed. How is the training effected due to 1 MC sample?
>
> We use a large number of MC samples in order to reduce the variance of the estimation (e.g. 1000 samples for every CIFAR10 validation sample). However, for a larger dataset (as Imagenet) evaluating  thousand MC samples for a single validation sample lasts too long. Hence we decrease the number of MC samples in order to obtain the estimation in less time. The decrease in the number of MC samples is compensated with the dataset size, so the dataset likelihood estimation still has a small variance. We will further elaborate per-sample estimation variance with respect to the number of MC samples.
> We did not observe any sign of poor training due to using a single MC sample. Still, we use more than one MC sample during the evaluation to obtain a better estimate of the likelihood.
>
> > The related work section seems to be minimal and should be expanded to include relevant literature on normalizing flows and explicit generative models in general. The related work of [1,2,3] can be referred for details.
>
> We will expand the related work in future revisions of this manuscript as suggested by the reviewer.

---

> > ### Comment · Reviewer_EGVd · 2021-08-26
> > **Post Rebuttal**
> >
> > Thank you for addressing the comments in the rebuttal. The concerns with respect to the architectural details are adequately addressed. The proposed architecture is claimed to be less memory intensive compared to that of Vflow and Flow++ . However, it is not clear if this does scale with the dimensionality of the dataset. Overall,  the manuscript needs work with respect to discussion of related work, details of the architecture and training. While the method is incremental, the memory footprint of the approach and the results are indeed of interest to the community. I am thus in favor of acceptance.

---

### Official Review · Reviewer_5hYR · 2021-07-18

**Rating:** 6
**Confidence:** 3

**Summary:**

This paper presents dense flow that are normalizing flows with additional latent variable after each transformation. This new architecture is tested on generative modelling of images and yield results close or better than concurrent methods whereas dense flows require lesss computational power for training and are relatively fast at generation time.


**Ethical Concerns:**

/

**Ethics Review Area:**

["I don’t know"]

**Limitations And Societal Impact:**

/

**Main Review:**

Originality:
The method suggested in this paper of adding latent variables to NFs has already been proposed. Authors say the difference with their methods is that they add latent after each transformation steps but I don’t believe this is significantly different from this other work. However this paper demonstrates those additional latent combined with attention mechanism and multi-scale like NF architectures are useful for reaching great generative performance.

Quality:
Experiments show the model is well performing in terms of likelihood scores, sample quality and computation speed. However results regarding speed are not benchmarked very seriously as methods are using very different hardware which makes it hard to have an exact feeling on where each method stands. In particular it is very difficult to assess the memory usage of different methods and how the proposed method stands with respect to others.
It is not clear why table 3 is only providing FID scores for CIFAR10 and not other Imagenet and Celeba.
I have also the following remarks:
• L47-49: I don’t agree with your claim about the expressivity of NFs. You can always increase the capacity of the neural networks used to do the transformation although those transformation are one to one.
• Some indices not correct in example 1.
• Section 2.2: You should maybe describe multi scale architectures.

Clarity:
Overall the paper is well written however I have the following remarks.
• 148-156: this is not very clear.

Significance:
It is not clear whether the proposed method is solving any existing problem as many methods exist for creating good generative models. In particular the complexity of the architecture is relatively high and it is not clear if the nice results achieved are really coming from any new idea or just a very nice combination of existing work. If it is only the latter I would expect to see stronger results.


**Time Spent Reviewing:**

5

---

> ### Author Response · Authors · 2021-08-10
> **Authors response to Reviewer 5hYR**
>
> We would like to thank the reviewer for constructive feedback.
>
> ###  Originality & significance
>
> We agree with the reviewer that augmenting the input with noise channels has been proposed in earlier work, as discussed in section 4. However, our manuscript presents two additional insights which make a considerable influence to the NF performance. Our first insight is that expressiveness of a NF does not only depend on network width but also on its distribution across the depth. Our second insight is that the expressiveness can also be improved by conditioning the noise with densely connected cross-unit coupling.
> A combination of these two insights and the proposed memory-efficient intra-unit couplings substantially reduces computational requirements for training normalizing flows. This result could promote usage of deep generative models in new downstream tasks.
>
> ### Quality
>
> > In particular it is very difficult to assess the memory usage of different methods and how the proposed method stands with respect to others.
>
> Thank you for detecting a possible inconvenience for the reader. For a more explicit comparison,  we will extend Table 2 with total memory consumption and present our V100 experiments for all datasets.
> We note that the original Table 2 provided information for the smallest GPU which allows the training of our model. We also note that the training time is approximately equal on all GPUs due to small resolution of training images.
>
> >It is not clear why table 3 is only providing FID scores for CIFAR10 and not other Imagenet and Celeba.
>
> Concurrent methods report FID score only for CIFAR10. We agree with the reviewer that believe the FID scores should be reported for various datasets. Our submission provides these metrics in L243-244.
>
> > L47-49: I don’t agree with your claim about the expressivity of NFs. You can always increase the capacity of the neural networks used to do the transformation although those transformation are one to one.
>
> We agree with the reviewer that our text in L46-49 is not very well written. We thank the reviewer and apologize for the inconvenience! We wished to argue that increasing the coupling complexity does not necessarily improve the recognition performance, as shown in [24]. Future revisions will clarify this issue.
>
> > Section 2.2: You should maybe describe multi scale architectures.
>
> Multi-scale normalizing flows [18,19] reduce the resolution of the latent representation by space-to-channel reshaping. Subsequently, they reduce the network width by factoring out a subset of dimensions which are evaluated directly. This is a useful suggestion. We shall include this discussion in the next revision of this manuscript.
>
> >  Some indices not correct in example 1.
>
> Thank you. Each occurrence of index i should be replaced by index 1 (eg e_i -> e_1).
>
> ### Clarity
>
> > 148-156: this is not very clear.
>
> We will revise the paragraph in order to more clearly describe intra-unit coupling, in accordance with Fig. 2. Intra-unit coupling consists of two main components: i) the densely connected convolutional branch which captures local correlations, and ii) the self-attention branch which captures global context. The outputs of these two branches are fused into a single tensor.

---

> > ### Comment · Reviewer_5hYR · 2021-08-22
> > **Post Rebuttal**
> >
> > I thank the reviewers for addressing most of the concerns I had. I will improve my score to 6. I am not going higher because I feel this work is "just" a very nice proof of work but does not introduce any strong idea or a surprising result. I feel this paper could be interesting to read for some practitioners but I am not sure it reaches the significance standards of neurips publications. Depending on inputs from other reviewers or the area chair I could of course change my mind!

---

### Decision · Program_Chairs · 2021-09-27

**Decision:**

Accept (Poster)

**Comment:**

The paper proposes a normalizing-flow architecture that includes latent variables and is geared towards image modelling. The paper demonstrates good empirical performance of the proposed architecture at modest computational budgets.

There is a clear consensus among reviewers about the strengths and weaknesses of the paper: the main strength is the good empirical performance of the proposed architecture; the main drawback is the incremental novelty. Overall, the paper seems a useful contribution for generative-modelling practitioners, so I'm happy to recommend acceptance.